# Hippo/YAP Signaling Pathway: A Promising Therapeutic Target in Bone Paediatric Cancers?

**DOI:** 10.3390/cancers12030645

**Published:** 2020-03-10

**Authors:** Sarah Morice, Geoffroy Danieau, Françoise Rédini, Bénédicte Brounais-Le-Royer, Franck Verrecchia

**Affiliations:** INSERM, UMR1238, Bone Sarcoma and Remodeling of Calcified Tissues, Nantes University, 44035 Nantes, France; sarah.morice@univ-nantes.fr (S.M.); geoffroy.danieau@univ-nantes.fr (G.D.); francoise.redini@univ-nantes.fr (F.R.); benedicte.brounais@univ-nantes.fr (B.B.-L.-R.)

**Keywords:** hippo signaling pathway, YAP, osteosarcoma, Ewing sarcoma

## Abstract

Osteosarcoma and Ewing sarcoma are the most prevalent bone pediatric tumors. Despite intensive basic and medical research studies to discover new therapeutics and to improve current treatments, almost 40% of osteosarcoma and Ewing sarcoma patients succumb to the disease. Patients with poor prognosis are related to either the presence of metastases at diagnosis or resistance to chemotherapy. Over the past ten years, considerable interest for the Hippo/YAP signaling pathway has taken place within the cancer research community. This signaling pathway operates at different steps of tumor progression: Primary tumor growth, angiogenesis, epithelial to mesenchymal transition, and metastatic dissemination. This review discusses the current knowledge about the involvement of the Hippo signaling pathway in cancer and specifically in paediatric bone sarcoma progression.

## 1. Introduction: First Discoveries about the Hippo Signaling Pathway

The Hippo signaling pathway was discovered at the end of the 20st century, when it was first described as a key regulator of tissue growth in Drosophila. In 1995, Noll and Bryant [1] in addition to Stewart and Yu [2] demonstrated aberrant and strong tissue growth in Drosophila in response to a loss of Wst (warts) protein expression. This was the start of many studies on the partner factors of the Hippo signaling pathway. In the early 2000s, Sav (salvador), Hippo, and Mob (monopolar spindle-one-binder) proteins were described [3,4,5]. A functional and biochemical characterization of the Salvador-Warts-Hippo signaling pathway was thus established [6,7]. This corresponds to a cascade of phosphorylation by protein kinases, in which Hpo phosphorylates and activates Wts, which in turn represses the transcription of target genes via a transcription inhibitor unknown at that time. After these studies, Yki (yorkie) was identified in 2003 by Pan and Coll and was defined as a transcription factor coactivator and as a direct target of Wts [8,9,10].

The Hippo signaling pathway is highly conserved among animal species. The 1990s saw discovery of homologues components of the Hippo signaling pathway in mammals such as YAP (yes-associated transcription factor coactivator), even before the functional characterization of the pathway in Drosophila [11]. Nevertheless, results obtained in Drosophila have been extended to mammals, outlining the Hippo signaling pathway described by Duojia Pan and Coll in 2007 [8,12].

A decade of intense research has extended the Hippo phosphorylation cascade into a complex signaling network that is linked to different extracellular signals such as cell adhesion, polarity or mechanical stress. Recent studies have further implicated the Hippo pathway in various physiological processes and other pathologies, such as the regulation of stem cell differentiation, tissue regeneration, immunity, or cancer.

## 2. Components of Hippo pathway in mammals

Schematically (Figure 1), the core component of the Hippo pathway is a cascade of kinases in which the mammalian MST1/2 (STE20-like kinase 1/2) protein phosphorylates and activates LATS1/2 (large tumor suppressor 1/2) protein [2,13]. The purpose of this kinase cascade is to restrict the activity of two transcriptional coactivators; YAP and TAZ (transcriptional coactivator with PDZ-binding motif). When YAP or YAZ are not phosphorylated, they translocate into the nucleus to bind transcription factors, including TEAD (transcriptional enhanced associate domain) proteins. This complex activates the expression of several genes involved in many cellular processes such as cell proliferation, survival, or migration [13,14,15,16].

This cascade of phosphorylation is initiated by the phosphorylation of MST1/2 on threonine 183/180, resulting in MST1/2 activation [17,18]. It has been demonstrated that MST1/2 activation can be achieved by auto phosphorylation and kinases such as TAO1. The MST1/2 protein forms a homodimer at its C-terminal domain: Sav–Rassf–Hpo or SARAH domains. Each subunit of MST1/2 can activate the other subunit by phosphorylating the activation loop itself. The dimerization of MST1/2 is modulated by two other proteins of the SARAH complex: SAV1 and RASSF. SAV1 promotes self-activation of MST1/2, unlike the proteins of the RASSF family which, by forming heterodimers with MST1/2, prevents its activation [11,19,20]. 

The active MST1/2 protein phosphorylates SAV1 and MO1A/B (MOB kinase activator 1A and 1B), which are two scaffold proteins. The exact role of SAV1 is still poorly described. It has been suggested that SAV1 may facilitate the interaction between MST1/2 and LATS1/2 or may recruit MST1/2 to the cell membrane. MO1A/B is better described. It promotes signaling by facilitating the kinase activity of LATS1/2 and the phosphorylation of YAP/TAZ [21,22]. 

Another key player in this cascade of phosphorylation is NF2 (neurofibromatosis type 2), which directly interacts with LATS1/2 and thus facilitates its phosphorylation by the MST1/2-SAV1 complex. In turn, active LATS1/2 phosphorylates YAP and TAZ on the serine S127 and S381, resulting in their inactivation. Transcription coactivators are sequestered in cytoplasm (14-3-3 binds) and forms a complex, which leads to proteasomal degradation. When the Hippo pathway is weakly active, YAP translocates to the nucleus and leads to increased target genes expression such as CTGF (connective tissue growth factor) or CYR61 (cysteine-rich angiogenic inducer 61). However, since YAP does not have an intrinsic DNA-binding domain, transcription factors are required to mediate transcriptional activity [18,23,24]. 

## 3. YAP and Solid Cancers

Given the crucial role of the Hippo pathway in the regulation of organ size and development, it is not surprising that dysfunctions involving this signaling pathway lead to the development of cancers.

Over the last decade, many authors have demonstrated the involvement of YAP/TAZ during carcinogenesis and overall tumor growth. Schematically, tumor cells use the biological properties of YAP/TAZ to promote their ability to proliferate, migrate, and invade. That ability also facilitates tumor formation and progression. Immunohistochemical analyses of the biopsies of many human cancers suggest an increase of YAP/TAZ activity compared to healthy tissues. In this context, meta-analyses of more than 20 studies on more than 9000 tumors have indicated an activation of YAP/TAZ signaling in ovarian, head, and neck cancers, for example. Gastro-intestinal and gynecological cancers are more often associated with mutations in LATS1/2 and NF2 inhibitory kinases [3,25,26]. 

Overexpression of YAP, which is associated with a high level of TEAD, correlates with poor prognosis and increases the resistance to chemotherapy in different cancers [27]. However, the mechanisms that underlie the amplification of YAP/TAZ or TEAD expression are not yet known. Recent studies have suggested that there is cooperation between the Hippo pathway and other factors such as chromatin remodeling or other signaling pathways. In 2017, Saladi and Coll demonstrated that amplification of ACTL6A (actin-like protein 6A) and p63 increases the expression of YAP. Other signaling pathways can interact with the Hippo/YAP/TAZ pathway [28]. Their involvement in tumor development is often associated with the inhibition of YAP/TAZ inhibitory kinases, notably PI3K (phosphoinositide 3-kinase), which inhibits LATS1/2. A loss of regulator activity upstream the Hippo pathway may be responsible for the nuclear translocation of YAP to promote tumor cells proliferation. YAP nuclear localization is strongly associated with NF2 tumor suppressor mutations in nervous system cancers. In addition, genetic alterations and methylations on LATS1/2 tumor suppressors have been observed in various cancers [29,30]. More rarely, deregulation of the Hippo pathway involves the formation of fusion proteins, notably TAZ-CAMTA1 fusion protein, which is found in 90% of vascular cancers. This fusion protein is constitutively active and is not regulated by Hippo pathway components [31].

### 3.1. Primary Tumor Growth

The Hippo signaling pathway, first defined as a regulatory pathway that controls cell proliferation and organ size, is widely involved in primary tumor growth. Many studies have described the role of the YAP-associated transcription factor TEAD in this phenomenon. TEAD in association with YAP leads to the transcription of target genes known to be involved in cell proliferation and tissue homeostasis. These genes include CYR61 and CTGF, Wnt5A, DKK1 (dickkopf WNT signaling pathway inhibitor 1), TGFB2 (transforming growth factor beta-2), MYC, etc. [26,32,33].

The role of the Hippo pathway in the development of cancers is not limited to direct action on tumor cells. It also affects their microenvironment, such as their vascular microenvironment.

### 3.2. Angiogenesis

Angiogenesis may be defined as the formation of new blood vessels from pre-existing vascularization. 

Since angiogenesis is a highly regulated process during development, it is not surprising that an imbalance in new vessel formation can lead to different pathologies, including cancers. Many studies have demonstrated the impact of primary tumor growth and metastasis formation on neovascularization. Usually, when a tumor reaches a size of about 2 mm in diameter, it can no longer grow without the nutrients provided by neovascularization. Blood vessels are essential for tumor growth and allow tumor cells to metastasize from the primary tumor. Tumor cells that reach the bloodstream migrate and niche into other organs, which in turn induce new blood vessels formation. Many studies have suggested that tumor angiogenesis is induced when primary tumor growth causes an imbalance of the ratio between pro-angiogenic and anti-angiogenic compounds. Increasing the size of the primary tumor reduces oxygen exchange between tumor cells and blood, resulting in the activation of hypoxic pathways. HIF (hypoxia-inducible factor) expression is increased, resulting in the overexpression of pro-angiogenic molecules such as VEGF (vascular endothelial growth factor), FGF (fibroblast growth factor), and MMPs (matrix metalloproteinases) [34,35,36].

The cytokines VEGFs are well-known pro-angiogenic factors. Schematically, VEGFs can activate signaling pathways involved in angiogenesis, for example by increasing the production of factors that increase endothelial cell proliferation such as MMPS. It appears that the upstream signals of pro-angiogenic factors are redundant with other signaling pathways such as those of FGFR (fibroblast growth factor receptor) or PDGFR (platelet-derived growth factor receptor). Endothelial cells are sensitive to two main factors: Hypoxia-inducible factor 1 (HIF-1), which is expressed when the oxygen level is very low, and MMPs that cause ECM (extracellular matrix) degradation and release many growth factors that impact endothelial cell migration. During the early stages of angiogenesis, endothelial cells lose their junctions with adjacent cells and change their shape to increase their motility. In 2000, Glienke and Coll demonstrated the ability of endothelial cells to form neo-vessels in cultured matrigel [37]. They compared these tube-forming endothelial cells to naive endothelial cells to demonstrate the key role of many compounds known to be transcriptional targets of YAP, including CTGF, CYR61, and AMOT2 (angiomantin 2). Other research teams have therefore carried on studying the roles of the Hippo pathway in angiogenesis, starting with proteins belonging to the AMOT family: AMOT, AMOTL1, AMOTL2. These proteins regulate endothelial cell motility and are both known to interact with YAP and to be transcriptional targets of YAP/TEAD complex [38,39,40,41,42].

The permeability and integrity of blood vessels are also regulated during angiogenesis due to the CD44 marker located on endothelial cell surfaces [43]. CD44 can regulate the level of expression and activity of MMPs. One study has associated CD44 marker with NF2, which is a regulator of the Hippo pathway described above [44]. However, the consequences of CD44–Hippo interactions on YAP/TAZ effectors are not yet fully understood. 

The microenvironment of endothelial cells is an essential element in their regulation, notably via ECM’s physical properties. High matrix stiffness normally leads to YAP cytoplasmic localization in normal conditions [45,46]. Endothelial cells are sensitive to matrix stiffness modification. In 2017, Mochizuki and Coll demonstrated that when endothelial cells are in a restricted space, YAP is cytoplasmic. However, when cells are stretched, YAP is nuclear and leads to endothelial cell proliferation [47,48]. 

Although the main signaling cascade that regulates angiogenesis is the VEGF/VEGFR pathway, others can also be involved. For example, the TGF-β signaling pathway regulates endothelial cell proliferation, differentiation, and migration [49,50,51]. TGF-β cytokines promote the gene transcription involved in angiogenesis mainly via its transcription factor Smad3. Numerous in vivo and in vitro studies have thus demonstrated the importance of TGF-β in vessel formation. Interestingly, many studies have demonstrated the functional interaction between Hippo/YAP and TGF-β signaling pathways [52,53].

### 3.3. Metastatic Dissemination

Metastatic dissemination can be defined as the ability of tumor cells to migrate from the primary site to distant sites. Metastatic dissemination is thus a critical step in cancer treatment and is associated with a poor prognosis. Several studies have highlighted the Hippo pathway’s involvement in various cancers, which led scientists to study its implication in metastasis development. A series of studies conducted over the last 10 years has highlighted the importance of YAP in various molecular mechanisms associated with metastasis dissemination.

In the late phases of tumorigenesis, tumor cells undergo modifications under the influence of the microenvironment, resulting in an EMT (epithelial-mesenchymal transition) process. EMT is an essential process for metastatic dissemination and involves epithelial cells losing their polarity and their inter-cellular junction. During this process, cells acquire mesenchymal phenotypes, allowing tumor cells to migrate, invade, and thus metastasize. These EMT events are initiated by the inhibition of E-cadherin in response to signaling pathways such as the Wnt pathway and the TGF-β pathway. These events are followed by increasing expression of mesenchymal proteins such as vimentin or fibronectin, which is associated with the increasing expression of transcriptional factors such as snail, slug, or twist. The loss of E-cadherin is associated with TGF-β activity in several cancers, although several studies have demonstrated that other signaling pathways such as Wnt, sonic hedgehog, or the Hippo signaling pathway cooperate with the TGF-β pathway during this process of EMT. Notably, EMT is a reversible process; tumor cells can recover an epithelial phenotype that is more appropriate to the development of secondary tumors [54,55]. 

Few studies have sought to explain how YAP induces EMT. Li and Coll have reported that TAZ overexpression in mammary epithelial cells induces SOX2 (SRY-related HMG-box-2) expression and initiates EMT [56]. Overholzer and Coll have demonstrated that YAP overexpression in mammary epithelial cells results in a change of cell conformation associated with an expression profile related to EMT [57]. This publication was followed by others with similar findings on cholangiocarcinoma cells, mouse breast cells, pancreatic cancer cells, and other types of cancers [58]. TAZ leads to the overexpression of essential EMT transcriptional factors such as snail, slug, twist, ZEB1 (zinc finger e-box binding homeobox 1), and FOX2 with increased expression of vimentin, N-cadherins, and MMPs [58].

In addition, ZEB1 may interact directly with YAP to regulate the expression of target genes involved in EMT, associated with poor prognosis and high risk of relapse1. Similarly, Snail and Slug form a complex with YAP to regulate the differentiation and division of skeletal stem cells [59]. KRAS (Kirsten rat sarcoma viral oncogene homolog) stimulates the Fos-mediated transcriptional activity of YAP in a mouse model of lung cancer and activates the expression of genes involved in the EMT process [60]. Finally, a recent study has suggested a molecular mechanism by which YAP controls EMT: First by suppressing the expression of E-cadherin through a WT1-dependent mechanism, then by increasing Rac1 activity and promoting cell migration [61]. Significantly, YAP expression and activity are regulated by the EMT, suggesting a feedback loop between EMT and YAP activity.

Following EMT, tumor cells acquire an elongated morphology and the ability to migrate and invade. Deregulation of the Hippo pathway has been repeatedly associated with the regulation of these two major mechanisms in the formation of metastases, particularly in breast cancer, glioma, and colon cancer [62,63,64,65]. Most of the YAP/TAZ–TEAD transcriptional targets that potentially affect migration have been identified, including CYR61 and CTGF. Most published studies have implicated the TEAD transcription factor in the YAP-mediated migration process [66,67,68,69,70,71,72].

Intravasation is a crucial process that allows tumor cells to reach the bloodstream. Two recently published studies involved YAP in intravasation. The loss of LATS kinase activity induces intravasation of mammary tumor cells via YAP; the same results were found in uveal melanoma with a mutation on YAP activating GNAQ (G protein subunit alpha Q) [73,74]. However, the molecular mechanisms that may explain the role of YAP in intravasation have not yet been described. 

To survive in the blood circulation, tumor cells must resist mechanical stress, immune cell monitoring, and apoptosis caused by the loss of junctions between cells and ECM. The Hippo pathway is known to be involved in cell survival, especially because when YAP is active there is an inhibition of apoptosis via the repression of the BCL2-like1 (B-cell lymphoma 2-like1) protein and via the stimulation of the anti-apoptotic protein IAP (inhibitor of apoptosis) [75]. In 2012, Zhao and Coll suggested the molecular mechanism by which YAP controls tumor cell survival: The kinases LATS1/2 regulate YAP activation and thus its involvement in tumor cell survival [76]. 

Finally, extravasation allows tumor cells to leave the bloodstream to invade a secondary tissue. In this context, Sharif and Coll have demonstrated in mice and zebrafish that inhibition of YAP signaling decreases extravasation and colonization of secondary sites by mammary tumor cells [77].

All these studies suggest the role of YAP/TAZ in several steps during the metastatic process by inducing EMT, intravasation, and tumor cell survival. However, the molecular mechanisms are still poorly defined, and the level of scientific evidence in some studies remains weak. 

## 4. Hippo Signaling Pathway in Bone Pediatric Tumors

Primary bone tumors are rare cancers that can be divided into two categories: Benign primary bone tumors and malignant primary bone tumors, the latter of which includes osteosarcomas and Ewing’s sarcoma (EWs). Primary bone tumors represent less than 1% of cancers and about 10% of all childhood and young adult tumors (Figure 2). 

### 4.1. Ewing Sarcoma

Ewing’s sarcoma is the second most common primary malignant bone tumor in children and young adults after osteosarcoma. The annual incidence in the population is about three cases per million, with a slight prevalence in young boys at a ratio of 1.5. Approximately 85% of EWs occur in bone, with predominance in the diaphysis of long bones (femur, tibia, fibula, and humerus), but they can also be localized in the pelvic bones and chest cavity. Approximately 25% of patients show lung metastases at diagnosis. Ewing’s sarcoma is a very aggressive and osteolytic tumor that is characterized by rapid growth and massive destruction of affected bones, which can lead to bone pain and fractures [78]. The tumor’s immunohistochemical analysis indicates small round cells that exhibit neuronal markers such as NSE, S-100, and CD57. Ewing sarcoma’s cellular origin is not clearly established and remains contested, with two hypotheses that suggest an origin of either primary cells from the neural crest or MSC. Ewing’s sarcoma development is associated with a chromosomal translocation that results in a specific fusion gene between EWS and an ETS family gene. The translocation t(11;22)(q24;q12) between the FLI1 and EWS genes is the most common and occurs in 85% of cases, giving rise to the transcription factor EWS-FLI1. In about 10% of cases, a translocation between ERG and EWS, t(21;12)(22;12) is observed in about 10% of cases. The resultant EWS-FLI1 protein specifically recognizes FLI1’s DNA binding domain and modulates the expression of target genes involved in cell proliferation and metastatic dissemination [79]. 

There is scarce literature on the Hippo/YAP signaling pathway and EWs. Nevertheless, a first study on EWs tumor samples has demonstrated that YAP expression in EWs tumors did not correlate with patient survival [80]. In contrast, a more recent study conducted on a larger number of patients demonstrated that a high expression of YAP is associated with a poor prognosis, which suggests that the Hippo signaling pathway plays a key role in EWs progression. The study used the knockout of YAP1/TAZ in EWs cells to demonstrate that the activity of YAP1/TAZ drives the cells’ ability to proliferate and invade [81].

Regarding the role of EWS-FLI1 in regulating YAP signaling in EWs, He and Coll have demonstrated that EWS-FLI1 oncoprotein enhances tenascin expression by directly binding to its promoter region and that integrin α5β1-mediated YAP activation may be responsible for expression of YAP targets’ genes that are implicated in EWs tumor progression [82]. A molecular study has revealed that tenascin overexpression regulates YAP nuclear localization by decreasing its phosphorylation. Interestingly, one study has proven the role of the Hippo signaling pathway in the switch between a proliferative state and a migratory state of EWs cells associated with the fluctuation of EWS-FLI1 expression [83].

Furthermore, YAP seems to be able to regulate EWs progression via the regulation of oncogenes expression. For example, in vitro study has demonstrated that YAP induces the expression of BMI-1 oncogene in EWs, resulting in the loss of contact inhibition and high cell proliferation [84].

### 4.2. Osteosarcoma

Osteosarcoma (OS) is the most common primary bone tumor in children and young adults with an incidence of about four cases per million per year. This cancer accounts for about 35% of cases, followed by chondrosarcoma (25% of cases), and EWs (16% of cases). OS can occur in any bone but most commonly occurs in the metaphysis of long bones near the growth plate; epiphyses and diaphysis are rarely affected. The most common sites are the femur (48% of cases), tibia (27%), and humerus (15%). Less frequently, OS develops in the skull or facial bones (8% of cases) and pelvis (8%) [84,85,86]. Among pediatric cancers, OS ranks eighth after lymphomas and brain tumors. OS has two peaks of occurrence; the first during adolescence and the second in adulthood. OS remains very rare in children under five years of age at only 2% of cases. At diagnosis, about 20% of patients are present with metastases, most commonly in their lungs, but bone and lymph node metastases can also exist [87,88,89].

Literature on OS is less scarce than on EWs. Aberrations in Hippo signaling pathway were demonstrated in OS using immunochemistry approaches; this demonstrated a nuclear localization of YAP in OS patient tumor biopsies [90,91]. Interestingly, Bouvier and Coll have proven a correlation between YAP nuclear localization and a poor prognosis, allowing for the establishment of a vital prognostic at diagnosis [90].

The dysregulation of the Hippo signaling pathway seems to be associated with Sox-2 signaling in OS [92,93]. Basilico and Coll have revealed that Sox2 blocks the Hippo pathway by repressing the two Hippo activators—Nf2 (merlin) and WWC1 (kibra)—in OS. Repression of Nf2 and WWC1 stimulates YAP expression and enhances the tumorigenicity of OS. More recently, the same researcher team validated their results using Sox2 CKO animals. The crucial role of Sox-2 in driven OS progression was also demonstrated by using the PPARγ agonist thiazolidinedione (TZD) drugs. TZD affects OS cell proliferation only in the high SOX-2 expressing cancer cell population, by YAP sequestration in cytoplasm [92,93,94].

At transcriptional level, the transcriptional factor TEAD1 seems to be involved in YAP-driven OS development. Indeed, studies have used knockdown approaches to demonstrate the crucial role of TEAD in YAP-driven OS cell lines proliferation. The study’s use of knockdown approaches thus identifies the YAP1/TEAD1 transcriptional complex as the main dysregulated pathway of Hippo signaling in OS [95] (Figure 3). Interestingly, Hippo/YAP signaling was seen to interact with TGF-β signaling pathway in mesothelial or skin epithelial cells at transcriptional level [96,97] (Figure 3). Since several studies demonstrated the crucial role of the TGF-β to promote OS tumor progression [98,99,100], we could formulate the hypothesis that unless it has the ability to interact with TEAD, YAP would be able to interact with the TGF-β signaling pathway to promote OS development specifically metastatic process.

YAP expression in OS could be regulated by post-transcriptional or post-transductional modifications such as epigenetic or ubiquitination processes. In this context, recent studies have demonstrated the oncogenic role of various miRs in OS by increasing the expression and activity of YAP. For example, miR-375 increases the activity of YAP1, while miR-624-5p (which is overexpressed in OS) induces cell proliferation, migration, and invasion by increasing the amount of nuclear YAP [101,102]. Furthermore, a recent study has reported that FAT10, a ubiquitin-like protein, stabilizes YAP expression by regulating its ubiquitination and degradation [103].

TGF-β (transforming growth factor-β) dimers bind to TBRII receptor that induce the assembly of TBRI and TBRII receptors into a complex in which TBRII phosphorylates and activates TBRI. Smad3 is then phosphorylated and activated by TBRI. Activated Smad3 recruits Smad4 and this protein complex is translocated into the nucleus to regulate target gene expression in association with cofactors such as YAP. YAP thus could regulate the expression of specific targets involved in the cellular migration and thus in metastatic development. Smad7 is able to block this signaling pathway.

### 4.3. YAP-Signaling Inhibitors 

Considering the Hippo signaling pathway’s critical role in many pathologies, targeting that pathway seems to be an interesting therapeutic approach, particularly in oncology (Figure 4). Most inhibitors focus on the inhibition of YAP–TEAD interaction, but other compounds that inhibit YAP upstream regulators have been tested [104].

The most used compound is verteporfin, a benzoporphyrin-derived molecule already in clinical use in the treatment of age-related macular degeneration through photodynamic therapy. Liu-Chittenden and Coll have identified 71 compounds that inhibit TEAD activity in HEK293. Among these 71 components, three molecules belonging to the porphyrin family have demonstrated a strong ability to inhibit YAP–TEAD transcriptional activity. Verteporfin was selected among these three porphyrins to test the effects of YAP inhibition. Its efficacy has been established to inhibit liver growth after YAP overexpression [105]. 

Since the discovery of verteporfin in 2014, several studies have used it as a pharmacological inhibitor in vitro and in vivo. Verteporfin inhibits cell proliferation in vitro and in vivo in several carcinoma models, notably by inhibiting the expression of oncogenic factors such as c-myc or some cyclins [106,107]. Verteporfin decreases retinoblastoma cell cycling with an accumulation of cells in the early G0/G1 phase [108]. It is also able to increase cell death by increasing the expression of cleaved PARP-1 (poly [ADP-ribose] polymerase 1) or cleaved caspase-3, which are essential factors in the apoptotic death process [107]. One study reported verteporfin’s efficacy in re-sensitizing chemotherapy in resistant tumor cells [109]. At the transcriptional level, verteporfin decreases the expression of Hippo pathway target genes such as Cyr61 and CTGF in many cell types [108,110,111]. Most publications have focused on the inhibition of the YAP–TEAD interaction; however, few studies have demonstrated a decrease of YAP expression that is associated with cytoplasmic retention. This latest research also highlights the nonspecific effects of verteporfin [112,113]. Regarding OS, Zucchini and Coll demonstrated the ability of verteporfin to decrease YAP expression in SaOs2 OS cells and their ability to proliferate and to migrate [114].

Other compounds have also been identified, notably dasatinib, a tyrosine kinase inhibitor used in the treatment of certain types of chronic myeloid leukemia. Dasatinib inhibits Src kinase, which disrupts the JNK (Jun N-terminal kinase)-LIMD1 (LIM domains containing-1)-LATS cascade and therefore inhibits YAP nuclear translocation [115,116]. Currently, few studies have demonstrated the essential role of the Src-YAP axis in tumor development, but induction of YAP phosphorylation is a pharmacological approach that remains to be developed. Direct inhibition of YAP activity remains a challenge in cancer research. Recently, a new inhibitor of topoisomerases, A35, has produced some efficacy in YAP phosphorylation on serine 127, which blocks its nuclear translocation [117]. 

Piccolo and Coll have recently demonstrated the interaction between YAP and BRD4 (bromodomain-containing protein 4), a protein of the BET (bromo- and extra-terminal domain) family. BRD4 binds to acetylated histones and recruits transcription factors to DNA. JQ1 is a competitive inhibitor of BRD4 and decreases the amount of YAP-dependent transcribed genes [118]. In this context, Lamoureux and Coll, demonstrated that JQ1 significantly delays tumor growth in MNNG/HOS osteosarcoma xenograft and POS-1 sarcoma syngeneic models and prolongs cancer-specific survival [119].

The Hippo signaling pathway represents a real opportunity in cancer treatment. YAP activation and overexpression are associated with both tumor cell development and cancer progression. Unfortunately, there is no clinically available drug that targets the Hippo pathway. Identifying new drugs that specifically targets the Hippo pathway, including YAP, remains a challenge for pharmaceutical companies.

## 5. Conclusions and Future Direction

Accumulating evidence has demonstrated the rationale for targeting the Hippo signaling pathway in EWs and OS. Targeting the Hippo signaling pathway could affect both the primary growth tumor and the metastatic process. However, despite many improvements in OS and EWs treatments since the 1970s, resistance to chemotherapy remains a major and unsolved problem that prevents the total remission of some patients. Multiple cell survival mechanisms prevent current treatments from being fully effective. In addition, responses to chemotherapy agents differ from one patient to another due to OS and EWs tumor heterogeneity. Schematically, two types of resistance mechanisms to chemotherapy can be described: Innate resistance that is intrinsic to the cell and acquired resistance that appears after treatment. Different mechanisms may be involved in cells-resistance acquisition, such as ABC transporters, amplification of therapeutic targets, appearance of mutations on therapeutic targets, or activation of alternative survival signaling pathways [120,121,122]. Understanding and deciphering the molecular mechanisms underlying this cells-resistance process is essential for adapting treatments and preventing relapse. In this context, several studies involve YAP activation in resistance to chemotherapies, radiotherapies, and immunotherapies in many cancers. Interestingly, regarding OS, Wang and Coll have demonstrated YAP-regulated chemoresistance in MG63 osteosarcoma cells [123]. Further studies should be performed to improve the specific role of Hippo/YAP in OS and EWs tumor chemoresistance.

## Figures and Tables

**Figure 1 cancers-12-00645-f001:**
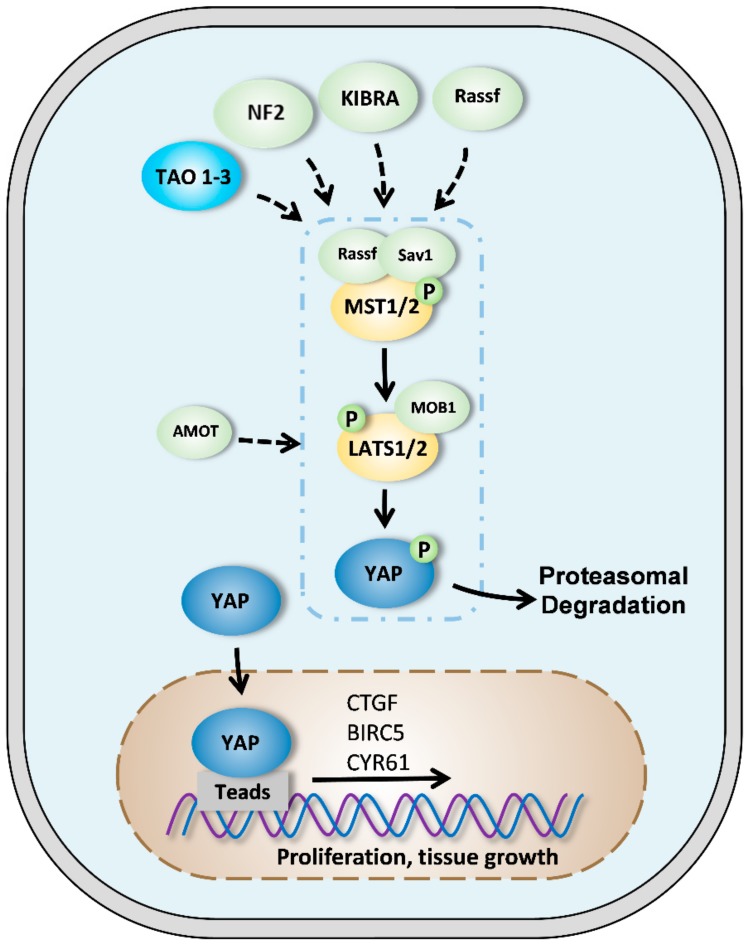
The Hippo/yes-associated protein (YAP) signaling pathway in mammals. When the Hippo signaling pathway is active, MST1/2 protein kinases (mammalian STE20-like kinase 1/2) are phosphorylated by NF2 (neurofibromatosis type 2), KIBRA, or TAO1-3. MST1/2 activates LATS1/2 (large tumor suppressor 1/2) proteins which are also stimulated by Sav1 (salvador) and Rassf (ras association domain family) proteins. LATS1/2 then phosphorylates YAP protein which is retained in the cytoplasm or is degraded by the proteasome. MOB1 (monopolar spindle-one-binder) and AMOT (angiomantin) proteins favor LATS1/2 phosphorylation and activity. When the Hippo signaling pathway is inactive, YAP is not phosphorylated and translocates to the nucleus where it can exert its transcriptional activity by binding to TEAD (transcriptional enhanced associate domain). YAP thus regulates the expression of specific targets such as CTGF (connective tissue growth factor), BIRC5 (baculoviral inhibitor of apoptosis repeat-containing 5), or Cyr61 (cysteine-rich angiogenic inducer 61).

**Figure 2 cancers-12-00645-f002:**
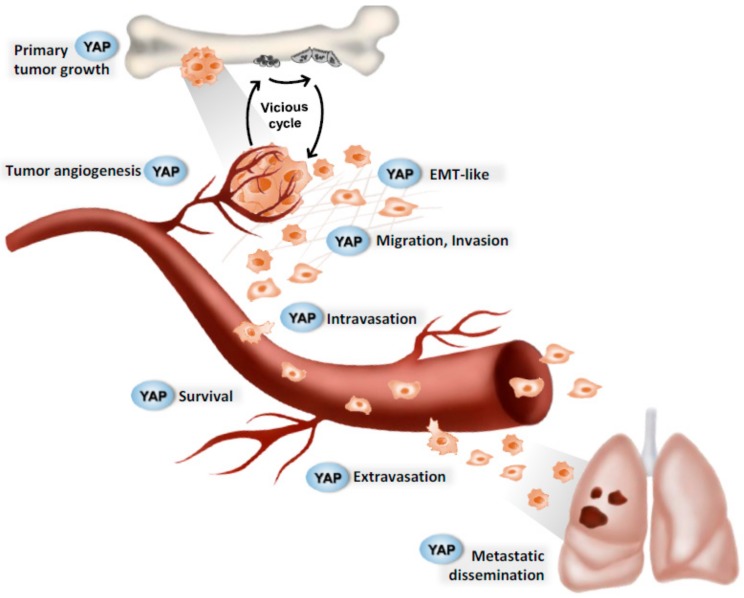
Successive steps in paediatric bone tumour progression. Hypothetical implication of the YAP signaling pathway. YAP (yes-associated transcription factor coactivator) plays a major role at various crucial steps during tumor progression. In pediatric bone tumors, the Hippo signaling pathway may be involved in primary tumor growth, tumor angiogenesis, epithelial-mesenchymal transition, intravasation, extravasation, cell survival, and metastatic dissemination.

**Figure 3 cancers-12-00645-f003:**
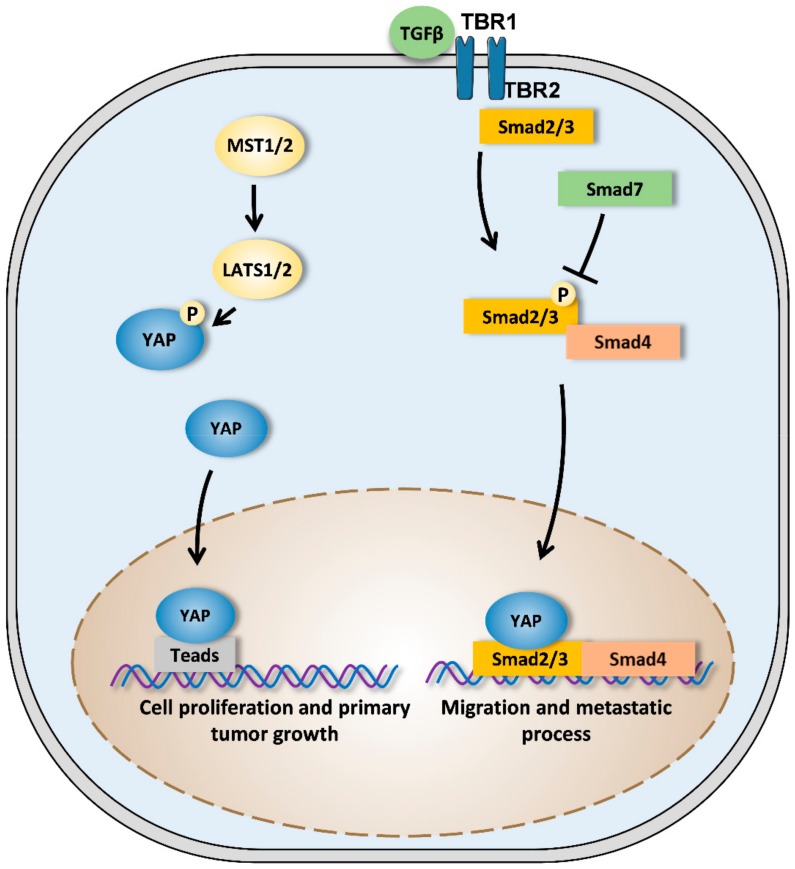
Crosstalk between YAP signaling pathway and both TEAD and TGF-β pathways. When the Hippo signaling pathway is inactive, MST1/2 (mammalian STE20-like kinase 1/2) activates LATS1/2 (large tumor suppressor 1/2) proteins. LATS1/2 then phosphorylates YAP (yes-associated protein) which is retained in the cytoplasm or is degraded by the proteasome. When the Hippo signaling pathway is inactive, YAP is not phosphorylated and translocates to the nucleus where it can exert its transcriptional activity by binding to TEAD (transcriptional enhanced associate domain). YAP regulates the expression of specific targets involved in the cellular proliferation and thus in the primary tumor growth.

**Figure 4 cancers-12-00645-f004:**
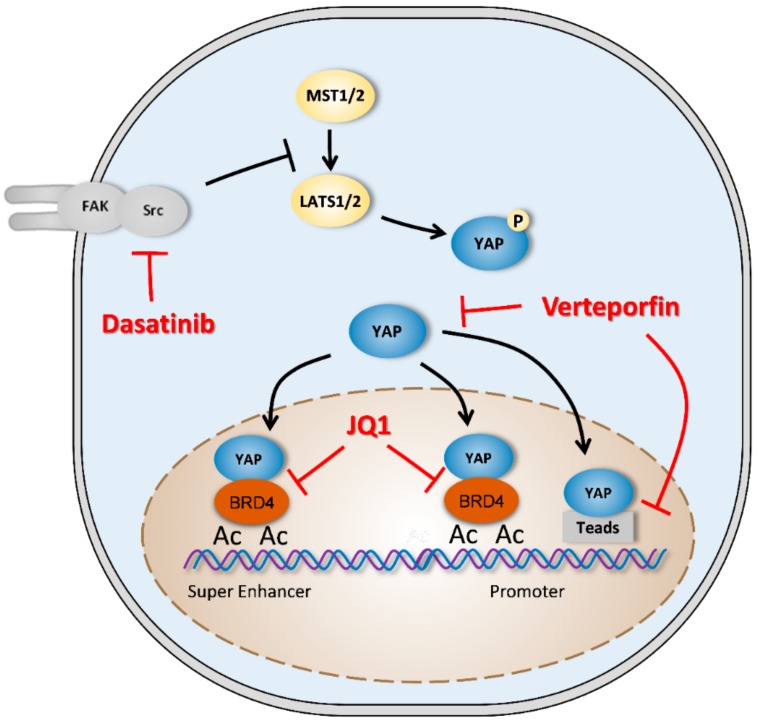
YAP drug targets. Verteporfin blocks YAP (yes-associated protein) signaling pathway either by decreasing YAP expression or by inhibiting YAP–TEAD (transcriptional enhanced associate domain) interactions. Dasatinid decreases LATS1/2 (large tumor suppressor 1/2) phosphorylation and activity driven by Src activity. JQ1 blocks YAP driven BRD4 (bromodomain-containing protein 4) activity by inhibiting BRD4 association to chromatin (Ac: Acetylations).

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
