# Peer review of "Hippo/YAP Signaling Pathway: A Promising Therapeutic Target in Bone Paediatric Cancers?"

_cancers, 2020, doi:10.3390/cancers12030645_

Round 1

Reviewer 1 Report

Manuscript # cancers-724861

Title: Hippo signaling pathway: a promising therapeutic target in bone pediatric cancers

Review:

Overall:

In this manuscript the authors review how Hippo signaling impacts pediatric tumors.  The review does a very good job of describing the relationship of Hippo to YAP and these complex signaling pathways/interaction and how they impact Osteosarcoma and Ewing sarcoma which are the most prevalent bone tumor types.  However, the addition of additional figures would greatly help to illustrate these complex relationships visually to enhance the overall quality of the manuscript.   

Minor Concerns:

Figure legends need to be clear and briefly defined by the legend. The abbreviations used in the figures need to be defined in the legends. Signaling Cascades are complex and the use of figures greatly improve understandability. Maybe the authors could add additional figures to illustrate the various Hippo and YAP affected pathways in tumor progression – ie., growth pathways, proliferation, EMT etc

Specific Comments:

Abstract – There needs to be more specific information on how the Hippo signaling pathway affects this type of cancer presented in the abstract as highlights. Figures— Overall quality of the images in terms of resolution must be improved as the images are ‘pixelated’ in the PDF. Signaling pathways are complex and the use of figures is justified in explaining them. However, the figure legend needs to define the abbreviations used. If the dotted box in Fig 1 is to represent “Hippo” this should be annotated in the actual figure and legend. Since YAP signaling is critical to how Hippo signaling regulates tumor progress, a figure that specifically illustrates how Hippo regulates YAP and then in general how YAP regulates tumor progression (YAP-targets) would be extremely useful for a readers’ understanding. A figure illustrating the described drug targets would also be useful. Figure 2. Title is Hippo signaling pathway…. There is no mention of Hippo in the actual figure ? The figure also needs to define the abbreviations similar to (b) above.

Author Response

Thank you for your comments about our manuscript entitled: “Hippo/YAP signaling pathway: a promising therapeutic target in bone paediatric cancers?”. We are very pleased with your comments that our paper is acceptable for publication pending minor revisions.

-) Two new figures have been added to improve understandability. The first describes the crosstalk between Hippo/YAP signaling both TEAD and TGF-b/Smad3 signaling pathways. Each crosstalk is correlated to the hypothetical role of these transcriptional interactions in the primary tumor growth and the metastatic process in osteosarcoma (new Figure 3). The second describes the YAP drug targets (new Figure 4).

-) Concerning the old Figure 2, the title has been modified to better illustrate the Figure (the term Hippo by the term YAP).

-) The abbreviations used in each figures have been now added in the figure legends.

In this context, please find herein the revised version of the manuscript. We hope that these corrections address the issues raised by you satisfactorilly, and that the attached revised version is acceptable for publication.

Reviewer 2 Report

An interesting topic concerning one of the important signaling pathways in cancer general, and , perhaps in paediatric cancers too.

While the topic is interesting, the actual reviewed text suffers from some minor issues:

  1. YAP-signaling inhibitors section should be placed after the sections on Ewing sarcoma and Osteosarcoma. Also, since the title of review focused on Hippo signaling and the mentioned tumors, authors should also extend/modify this section by providing evidence of whether YAP signaling inhibitors have already been tested in thse tumors.
  2. The current form of the text makes is lss attractive for reading, in part due to the layering of the txt as wel as the literary style. Some polishing is required here with booth language check and proofreading as well as the stylistical changes.
  3. Abbreviations have to be consistently explained the first time they are used. Also, since there is a consideble number of abbreviated names of various signals and factors authors must employ the unified system of their use - either explain the full name with all or avoid it altogether. 
  4. What role is played by TAZ? What is the relationship between TAZ and YAZ and YAP?

Author Response

Thank you for your comments about our manuscript entitled: “Hippo/YAP signaling pathway: a promising therapeutic target in bone paediatric cancers?”. We are very pleased with your comments that our paper is acceptable for publication pending minor revisions.

-) YAP-signaling inhibitors section has been placed after the sections on Ewing sarcoma and osteosarcoma.

-) To improve the attractive for reading, two new figures have been added. The first describes the crosstalk between Hippo/YAP signaling both TEAD and TGF-b/Smad3 signaling pathways. Each crosstalk is correlated to the hypothetical role of these transcriptional interactions in the primary tumor growth and the metastatic process in osteosarcoma (new Figure 3). The second describes the YAP drug targets (new Figure 4). In addition, language has been checked.

-) The abbreviations used in the text have been explained along the text and in the figure legends.

-) Since there are only little data regarding the role of TAZ in primary bone tumor development, we have chosen to focus this review solely on the role of YAP. Changes have been made based on this point

In this context, please find herein the revised version of the manuscript. We hope that these corrections address the issues raised by you satisfactorily, and that the attached revised version is acceptable for publication.